# Pre-plaque conformational changes in Alzheimer's disease-linked Aβ and APP

O. Klementieva[1], K. Willén[1], I. Martinsson[1], B. Israelsson[1], A. Engdahl[2], J. Cladera[3], P. Uvdal[2,4] & G.K. Gouras[1]

Reducing levels of the aggregation-prone Aβ peptide that accumulates in the brain with Alzheimer's disease (AD) has been a major target of experimental therapies. An alternative approach may be to stabilize the physiological conformation of Aβ. To date, the physiological state of Aβ in brain remains unclear, since the available methods used to process brain tissue for determination of Aβ aggregate conformation can in themselves alter the structure and/or composition of the aggregates. Here, using synchrotron-based Fourier transform infrared micro-spectroscopy, non-denaturing gel electrophoresis and conformational specific antibodies we show that the physiological conformations of Aβ and amyloid precursor protein (APP) in brain of transgenic mouse models of AD are altered before formation of amyloid plaques. Furthermore, focal Aβ aggregates in brain that precede amyloid plaque formation localize to synaptic terminals. These changes in the states of Aβ and APP that occur prior to plaque formation may provide novel targets for AD therapy.

[1] Experimental Dementia Research Unit, Department of Experimental Medical Science, Lund University, 22184 Lund, Sweden. [2] MAX IV Laboratory, Lund University, 22100 Lund, Sweden. [3] Department of Biochemistry and Molecular Biology, Universitat Autònoma de Barcelona, 08193 Bellaterra, Spain. [4] Chemical Physics, Department of Chemistry, Lund University, 22100 Lund, Sweden. Correspondence and requests for materials should be addressed to O.K. (email: oxana.klementieva@med.lu.se) or to G.K.G. (email: gunnar.gouras@med.lu.se).

Studies on the aggregation state of β-amyloid peptide (Aβ) in brain typically have relied on brain tissue being processed by techniques including homogenization, high-speed centrifugation and/or enrichment, all of which may themselves trigger alterations in protein structure and state of assembly of aggregation-prone proteins. Moreover, assaying whole-brain homogenate can make it difficult to detect localized pathology, which may occur in the early stages of the disease, before gradually progressing through the brain over time. A central question in AD concerns the mechanism by which Aβ structure contributes to neuropathology[1]. The up to 42/43 amino acid long Aβ peptides, cleaved from within the larger amyloid precursor protein (APP), are generally thought to normally exist in brain as monomers that then with age abnormally aggregate to insoluble fibrils in AD brain. Increasingly, soluble oligomers have been viewed as the neurotoxic form of Aβ. Various structures of Aβ oligomers are seen as important in AD, including particularly dimers, trimers, dodecamers and larger amyloid beta-derived diffusible ligands. However, studies on Aβ in brain tissue have generally relied on standard SDS gel electrophoresis or other methods, which can alter native protein conformations. Thus, at present, no consensus exists regarding the structures of Aβ species in brain associated with disease pathogenesis.

Here, to analyse the early changes in Aβ structure in brains of AD transgenic mouse models of β-amyloidosis, we used the non-destructive techniques of synchrotron-based two-dimensional Fourier transform infrared micro-spectroscopy imaging (μFTIR) and blue native polyacrylamide gel electrophoresis (BN-PAGE) complemented with 3D confocal immunofluorescence microscopy.

BN-PAGE is a separation method that is sensitive to alterations in the conformation of a protein and is therefore useful for the study of protein aggregation. Combined with western blot it provides higher sensitivity to detect Aβ than gel filtration or sucrose density ultra-centrifugation. μFTIR detects vibrations of main-chain carbonyl groups that occur in the wavenumber range of 1,600 to 1,700 cm$^{-1}$ (Amide I region)[2], allowing for the detection of specifically β-sheet structures[3]. Importantly, no tissue processing is required, and non-volatile tissue components that could be affected or lost during chemical processing remain *in situ* and contribute to the infrared spectrum[2–4]. μFTIR permits for the acquisition of spectra from samples as low as 100 pg (ref. 5); however, μFTIR is a non-destructive technique, since mid-IR photons are too low in energy (0.05–0.5 eV) to either break chemical bonds or to cause ionization[6,7]. Using FTIR it was shown that amyloid fibrils and native β-sheet proteins produce different Amide I bands[8], as they differ by the structural variability of the residues constructing their sheets, the average number of strands per sheet, and twist angles. Different morphologies of purified amyloid fibrils and native β-sheet proteins were confirmed *in vitro* by cryo-electron microscopy[9]. Although *in vitro* and *in vivo* environments are of course different and ideal controls for the FTIR experiments of brain tissue are challenging, μFTIR has been used to specifically detect amyloid fibrils in tissue sections[10,11]. Here we provide evidence of localized β-sheet elevations in brain tissue of AD transgenic mice using μFTIR that precede amyloid plaque formation. In addition, using BN-PAGE we show the loss of a low molecular weight Aβ complex and emergence of higher weight Aβ and APP complexes in AD transgenic mouse brains that occur concomitant with this pre-plaque rise in β-sheet content by μFTIR.

## Results

**Pre-plaque β-sheet transition in transgenic AD models.** To focus on the development of amyloid pathology, we used young AD transgenic Tg19959 mice harbouring two familial AD mutations in the amyloid precursor protein (APP), which show initial amyloid plaques at or just before 3 months of age[12]. μFTIR spectral maps were recorded in chemically unprocessed brain sections of Tg19959 and wild-type mice at 1, 2 and 3 months of age (Fig. 1). We verified that the sample preparation used for our μFTIR imaging of cryo-dried brain tissue did not introduce artificial β-sheet formation, since change of β-sheet did not occur with cryo-drying of synthetic Aβ42 preparations (Supplementary Fig. 1). The increase in intensity of a band around 1,627 cm$^{-1}$ in the amide I region of the infrared spectra (red on the μFTIR maps; Fig. 1a) correlated with the age of Tg19959 mice and is indicative of increased β-sheet structures already developing between 1 and 2 months, which is before amyloid plaque formation as determined by thioflavin S (ThS) staining. To better resolve the peak positions for β-sheet structures at around 1,627 cm$^{-1}$ and the band centred at 1,656 cm$^{-1}$ (a frequency characteristic of α-helical structures), we performed a second derivative analysis[10,13]. The second derivative spectra of wild type and 1 month-old Tg animals displayed a peak at around 1,640 cm$^{-1}$, which together with the less intense bands above 1,675 cm$^{-1}$, can be assigned to the intra-molecular β-sheet structures of native proteins[10,11]. The second derivative spectra of AD transgenic animals are characterized by a peak at about 1,627 cm$^{-1}$, due to inter-molecular β-sheet structures. The inter-molecular β-sheet content increased only in AD transgenic and not in wild-type mouse brains at these ages (Fig. 1b–g). This type of intermolecular β-sheet structure has been described as amyloid aggregates in brain tissue samples analysed by μFTIR (refs 11,14). However, β-sheet content in brain sections of 1-month-old Tg19959 mice was similar to those of wild-type mice. Immunolabelling of post-fixed brain sections adjacent to the FTIR brain sections with a C-terminal specific Aβ42 antibody indicated age-related increases of Aβ42 in transgenic but not wild-type mouse brain tissue. Of note, Aβ42 labelling in 2 month-old Tg19959 mice was punctate, and therefore not consistent with pre-fibrillar diffuse plaques, whereas β-sheet structures detected by μFTIR appear over significantly wider areas of cortex (Supplementary Fig. 2a), which likely is to some extent due to the lower spatial resolution of μFTIR. To complement the μFTIR data, we used the conformation-specific antibody OC (ref. 15) and ThS, which are both considered indicators of fibrillar amyloid structures. However, the adjacent Tg19959 brain sections to those showing early β-sheet structures by FTIR did not show evidence of amyloid fibrils with antibody OC or ThS (Supplementary Figs 3 and 4). The absence of ThS and OC antibody labelling may be due to the low concentration of fibrils and/or dye specificity, that ThS and OC labelling do not detect all types of fibrils[16,17], or due to a non-fibrillar nature of the β-sheet structures.

FTIR measurements of a model compound (synthetic Aβ1-40 monomers) showed a peak at 1,640 cm$^{-1}$ (Fig. 2a), which corresponds to unstructured Aβ40 monomers in the experimental solution (buffered D$_2$O)[18]. This signature was not observed for Aβ40 oligomers and fibrils, suggesting that unstructured monomers were not present in the samples. Instead, Aβ40 oligomers show a peak at about 1,630 cm$^{-1}$, which indicates the presence of β-sheets[18]. However, the electron microscopy, small angle X-ray scattering and ThT spectroscopy data support the non-fibrillar nature of these Aβ40 oligomers: lack of thin (12–15 nm), elongated (100–1,000 nm) and branched structures, which are characteristics of fibrils[19] (Fig. 2b–d). These data corroborate previously published experimental data[18] and although the secondary structure of *in vitro*-generated Aβ oligomers cannot be a true prototype for oligomers that may form *in vivo* in the complex environment of the brain, these data support the hypothesis that the β-structures detected in brain

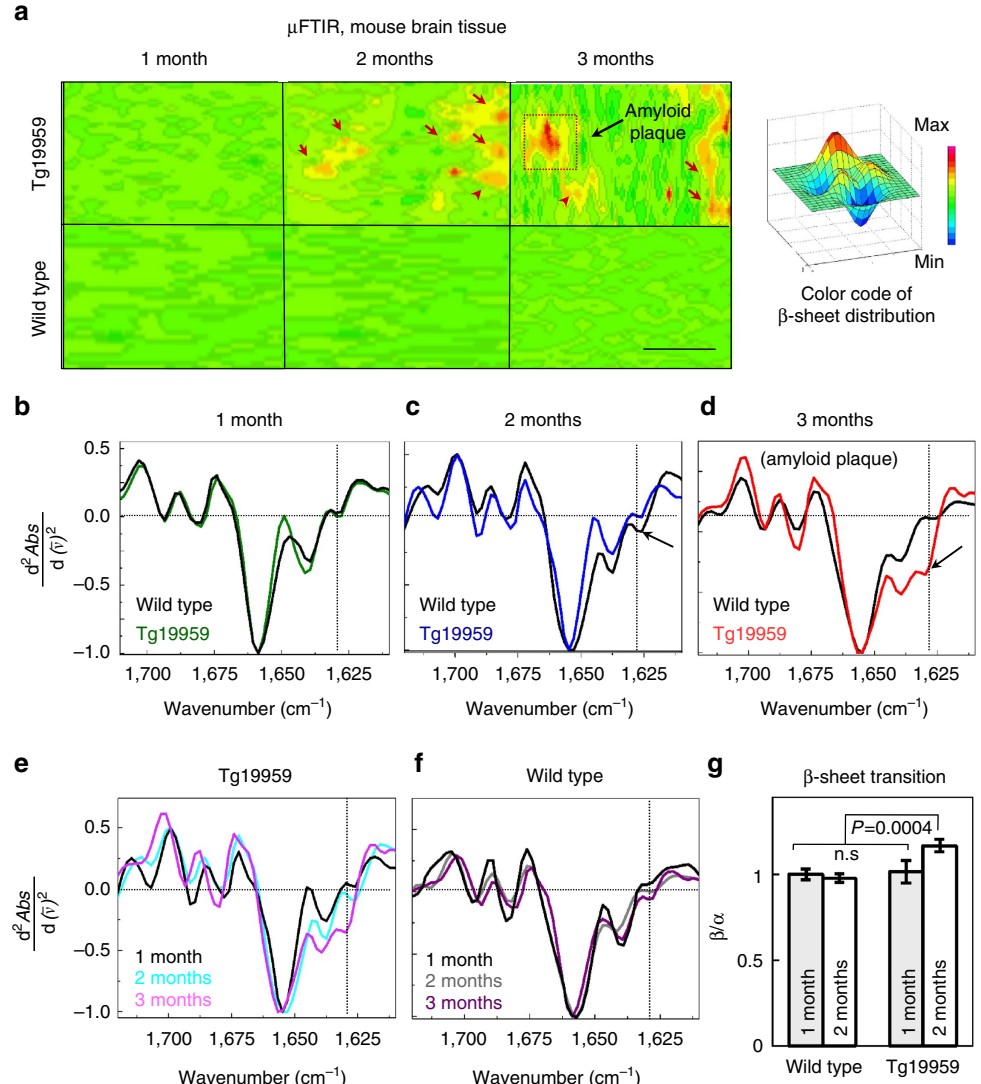

**Figure 1 | Early β-structural transition in transgenic AD mouse brains. (a)** FTIR maps were integrated for the β-sheet spectral region at 1,635–1,620 cm$^{-1}$ to visualize absorption intensities for the β-sheet content in brain sections of Tg19959 (upper panels) and wild type (lower panels) mice at 1, 2 and 3 months, respectively. β-sheet content is shown in red (arrows). Maps are representative; $N = 3$; N represents the number of animals per genotype/age. Scale bar, 50 μm. **(b)** Averaged and normalized 2nd derivatives of the Amide I absorption band; β-sheet structures in 1-month Tg19959 mice are similar to those in wild-type mice; averages were taken from 20 to 40 FTIR measurement positions per brain section and 3 brain sections per genotype/age). **(c)** Averaged and normalized 2nd derivatives of FTIR spectra taken from areas with increased β-sheet content in the corresponding μFTIR maps in **a**. The corresponding FTIR absorbance spectra with elevated β-sheet content are shown in Supplementary Fig. 2. The red arrow shows an increase in β-sheet content in Tg19959 mouse brain. Vertical dashed lines indicate the centre of the β-peaks. Horizontal dashed lines indicate ($x, y = 0$). **(d)** Same as in **c**, but for 3 month-old Tg19959 mice, with spectra taken from an area with high β-sheet content (indicated by red dotted square on the corresponding μFTIR map). **(e)** The overlap of the second derivatives corresponding to Tg19959 mice at different ages more clearly shows the progressive increase in β-content with age. **(f)** In comparison the overlap of the second derivatives corresponding to wild-type mice at different ages is similar. FTIR spectra are representative; FTIR spectra from different animals of the same age and genotype are shown in Supplementary Fig. 2c. **(g)** Statistical analysis of β-sheet content measured as the average of the ratio of peak intensities between 1,635 and 1,620 cm$^{-1}$ (β-sheet) to 1,656 cm$^{-1}$ (α-helix) in Tg19959 and wild-type mice as a function of age. Protein aggregation in wild-type mice is taken as 100%. ANOVA ($P < 0.01$) followed by Bonferroni's *post-hoc* comparisons test ($P < 0.01$). $N = 9$; N represents the number of brain sections. Data are represented as mean ± s.d.

tissue of Tg19959 mice at 2 months of age may originate from non-fibrillar (non-ThT and non-OC positive) β-sheet Aβ species formed in brain tissue prior to amyloid plaques.

To model the above findings in brain in a cellular system, we examined Aβ aggregation in AD transgenic neurons in culture, which show Aβ accumulation with time, similar to what occurs in brain with ageing[20]. μFTIR showed elevation of β-sheet structures in AD transgenic neurons between 12 and 19 days *in vitro*, compared to wild type neurons, respectively (Fig. 3a–c). Although age of neurons in culture[21] is not directly comparable to age of

neurons in brain, these results support the conclusion that β-sheet structures can be formed within neurons.

**Early changes in Aβ/APP complexes in AD transgenic mice.** To analyse changes in the native conformation of Aβ and APP in brain tissue BN-PAGE and subsequent western blot using different Aβ and/or APP specific antibodies was performed. For molecular weight estimation of native Aβ on BN-PAGE we used synthetic Aβ1-42, which runs as a mixture of monomers, dimers, trimers and tetramers. Interestingly, Aβ in brain of wild-type and

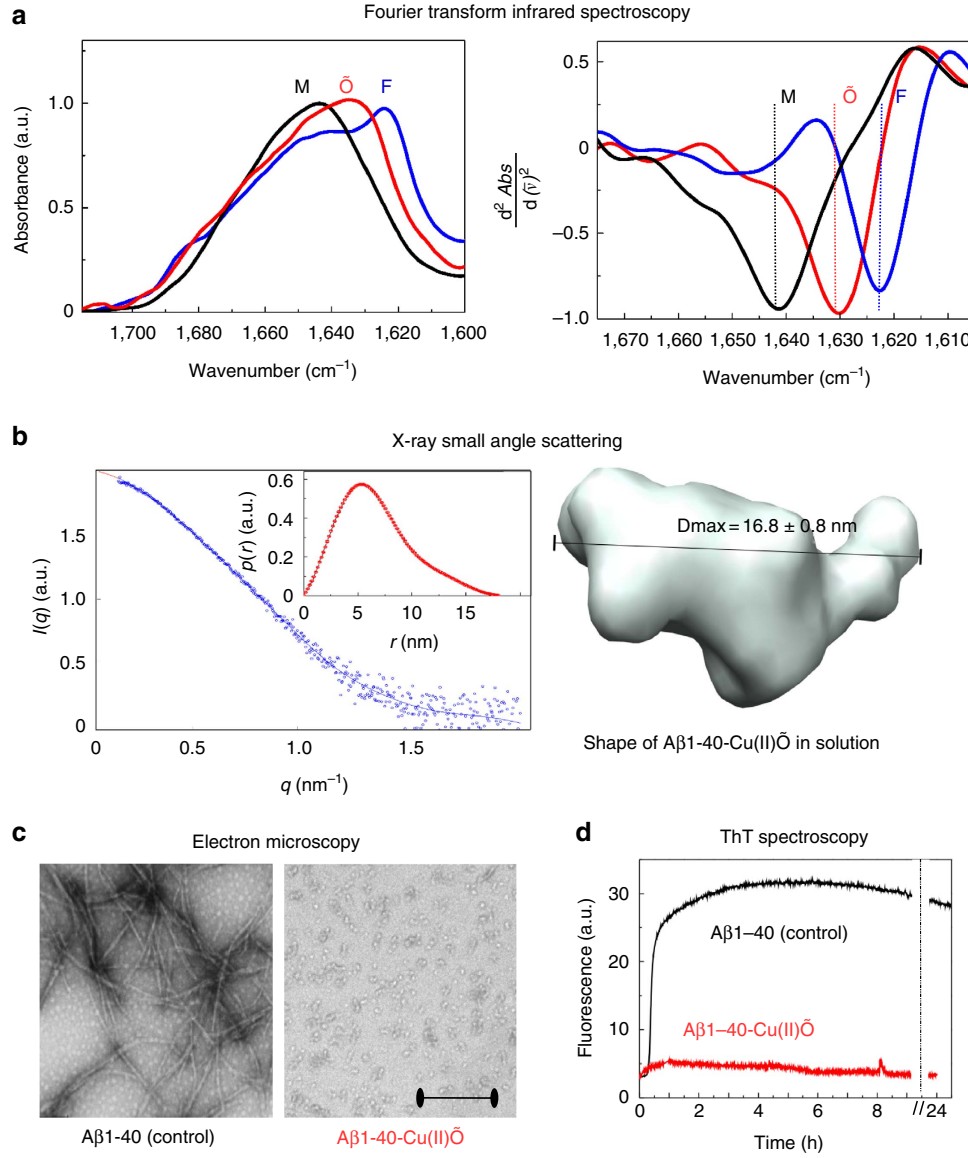

**Figure 2 | Non-fibrillar Aβ1-40 oligomers with β-sheet quaternary structure.** (**a**) Left panel: normalized FTIR spectra of Aβ1-40 monomers (M) in 10 mM Hepes/D₂O pD11, fibrils (F) after 24 h incubation in 10 mM Hepes pD7.4 and Aβ1-40-Cu(II)-induced oligomers (Aβ1-40-Cu(II)Õ) after 24 h incubation at 37 °C, pD7.4. Graphs show the Amide I region. Right panel: Second derivatives show a band centred at around 1,623 cm$^{-1}$ (blue, F) indicative of the existence of fibrillar β-structures. A broader band is centred around 1,643 cm$^{-1}$ (black, M), which can be assigned to a mixture of unordered and helical structures. A band centred at 1,630 cm$^{-1}$ (red, Õ) is indicative of the existence of β-sheet structure. (**b**) SAXS analysis of the Aβ1-40 Cu(II)Õ. In the graph: blue line shows the best fit for the model (spheres) calculated using GNOM (ref. 4); the blue circles correspond to the experimental data. Inset: pair distribution function p(r) with a single peak that corresponds to the quasi-globular shape with $D_{max} = 16.8 \pm 0.8$ nm. *De novo* three-dimensional reconstruction of the scattering entity of Aβ1-40-Cu(II)Õ using DAMMIN (ref. 4) after 10 independent DAMMIN reconstructions. A line indicates the maximum dimension ($D_{max}$). Scale bar, 200 nm. (**c**) Electron microscopy images show that after incubation for 24 h at 37 °C, Aβ1-40 (control) formed amyloid fibrils (right image) while Aβ1-40-Cu(II)Õ formed spherical aggregates. EM images are representative; 6–10 images were acquired for each EM grid. (**d**) Aβ1-40 aggregation was evaluated by ThT fluorescence; the change in ThT fluorescence showed that 25 μM Aβ1-40 Cu(II)Õ do not form fibrils (red), whereas a control sample, 25 μM Aβ1-40 alone, does form fibrils (black). ThT fluorescence was monitored during 24 h of incubation at 37 °C in 10 mM Hepes pH 7.4. The dashed line indicates a time point of sample collection for EM and FTIR. Kinetics are representative; at least three repeats were done.

1-month-old AD transgenic mice runs as a band closest to the 20 kDa low molecular weight marker on BN-PAGE (Fig. 4a,b; Supplementary Fig. 5a), as well as a higher molecular weight smear, which we show below, is however consistent with APP. This Aβ band corresponds to synthetic Aβ1-42 tetramers running on the same BN-PAGE (Fig. 4 and Supplementary Fig. 5). Remarkably, at and after 2 months of age there was a drop in the intensity of this Aβ band in AD transgenic mouse brains on BN-PAGE concomitant with an increase in Aβ aggregation

evident as a high molecular weight smear between 100–500 kDa. Blotting with antibody 369 against the C-terminus of APP/CTFs and 22C11 against the N-terminus of APP suggests that this Aβ tetramer band that drops at 2 months in AD transgenic mouse brain does not correspond to other APP fragments (Supplementary Fig. 5b). Overall, these results support the conclusion that under physiological conditions Aβ in brain initially exists as a low molecular weight protein complex consistent with Aβ tetramers. With the age-related increase in Aβ in AD

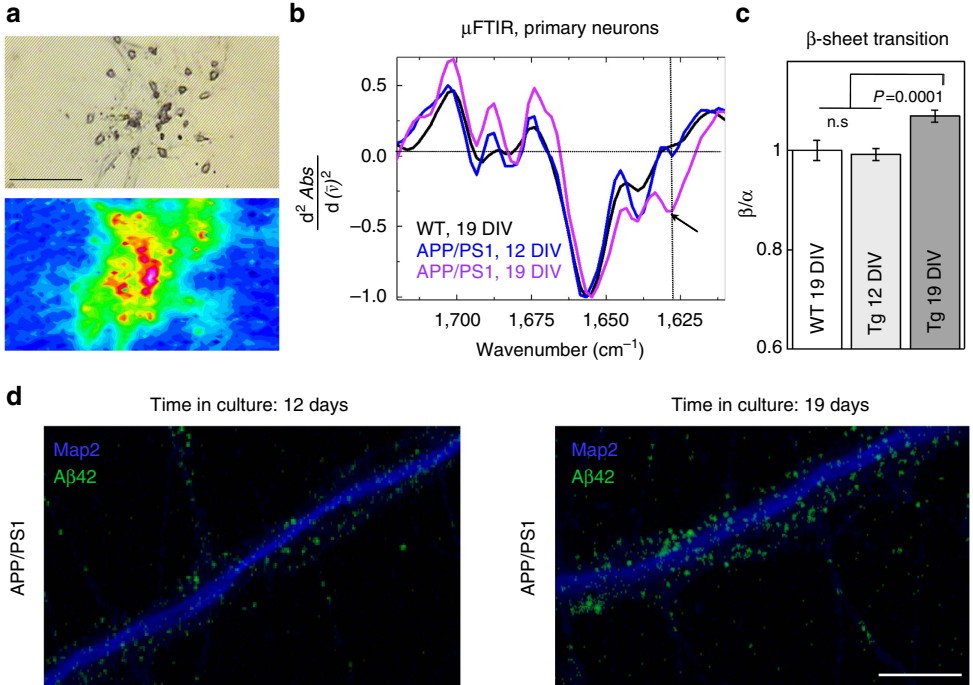

**Figure 3 | β-sheet aggregation in cultured AD transgenic primary neurons.** (**a**) Bright field image (upper panel) and FTIR map (lower panel) of AD neurons cultured for 19 DIV on a FTIR sample support. Scale bar, 50 µm. (**b**) Averaged and normalized second derivatives of FTIR spectra taken from APP/PS1 neurons at 12 and 19 DIV, and wild-type neurons at 19 DIV. Arrows indicate the β-sheet peak, which is evident only in APP/PS1 neurons at 19 DIV. (**c**) Statistical analysis of protein aggregation measured as the average of the protein aggregation ratios of 1,628 cm$^{-1}$ (β-structures) to 1,656 cm$^{-1}$ (α-structures) in AD transgenic and wild-type neurons as a function of days *in vitro*. β-sheet content in wild-type neurons is taken as 100%. ANOVA ($P < 0.01$) followed by Bonferroni's *post-hoc* comparisons test ($P < 0.01$); $N$ represents the number of neurons (50 cells per genotype/age). Data are represented as mean ± s.d. (**d**) High-resolution confocal microscopy images of immunofluorescently labelled Aβ42 (green) using antibody 12F4 and the dendritic marker MAP2 (blue) reveal increases in Aβ42 labelling between 12 and 19 DIV. Scale bar is 1 µm.

transgenic mice, this Aβ complex is then reduced as Aβ forms higher molecular weight aggregates. Strikingly, this change in the initial state of Aβ occurs concomitantly with the increase of β-structured content detected by µFTIR.

We next analysed the molecular weight of APP in brains by BN-PAGE (Fig. 4c; Supplementary Fig. 5c). Remarkably, the molecular weight patterns of APP also changed between 1 and 2 months in brains of Tg19959 mice before the appearance of amyloid plaques. Specifically, on BN-PAGE, APP in the membrane soluble fraction of Tg19959 mouse brains at 1 month of age runs similarly to APP in wild-type mouse brains as two bands, which appear to correspond to APP monomers and dimers[22]. In contrast, at 2 months of age, APP in Tg19959 mouse brains changes to three bands with molecular weights that correspond to about 140, 240 and 480 kDa. The appearance of the additional 480 kDa band supports that APP may aggregate as well. Interestingly, after 2 months of age, Tg19959 mouse brains show a smear with a molecular weight corresponding to 240–480 kDa on BN-PAGE using Aβ N-terminus antibody 82E1, which could suggest an interaction between Aβ and APP (ref. 23). Similar high-molecular weight Aβ was also observed on BN-PAGE in aged APP23 AD transgenic mice[24].

To investigate whether there is an age-dependent interaction between Aβ and APP, Tg19959 mouse brains were immuno-precipitated with the human-specific APP antibody P2-1 followed by dot blotting with the Aβ42 specific antibody 12F4, which indicated an age-dependent increase in the association of Aβ42 with APP (data not shown). To experimentally examine the interaction of Aβ42 with APP, transgenic and wild-type mouse neurons were treated with 1 µM synthetic human Aβ1-42 for 24 h. After treatment, BN-PAGE followed by Western blot with human Aβ/CTF antibody 82E1 and Aβ/APP antibody 6E10 revealed the added human Aβ as a band at 240 kDa, corresponding to the molecular weight of APP, only in the APP overexpressing AD transgenic neurons but not the wild-type neurons (Supplementary Fig. 6a). Furthermore, Aβ immuno-labelling of neurons treated with 1 µM synthetic Aβ1-42 showed a greater binding of Aβ1-42 to the cell surface of AD transgenic than to wild-type neurons (Supplementary Fig. 6b). Taken together these data support an interaction between Aβ and APP, and indicate that AD related protein aggregation is a complex process during which the physiological states of both Aβ and APP are affected.

**Aβ aggregation and synapse pathology.** To further define the localization of these pre-plaque β-aggregates and to evaluate potential pathological consequences of early Aβ-aggregation in brain, 1, 2 and 3 month-old Tg19959 mouse brain sections were labelled with antibodies directed against Aβ42 and synaptophysin or drebrin, markers of pre- and post-synaptic compartments, respectively. Three-dimensional confocal microscopy revealed that AD transgenic mouse brains show progressive Aβ42 accumulation in synaptic compartments (Fig. 5a; Supplementary Figs 7 and 8). In particular, post-synaptic Aβ42 labelling increased in AD transgenic mouse brains. In 1-month-old Tg19959 and wild-type mouse brains synaptic Aβ42 labelling is much less evident compared to older Tg19959 mice, consistent with prior immuno-electron microscopy of another AD trans-genic mouse model[25]. The increase of Aβ42 labelling at two months of age is concomitant with the FTIR and biochemical data showing increases in Aβ aggregation with age in these AD

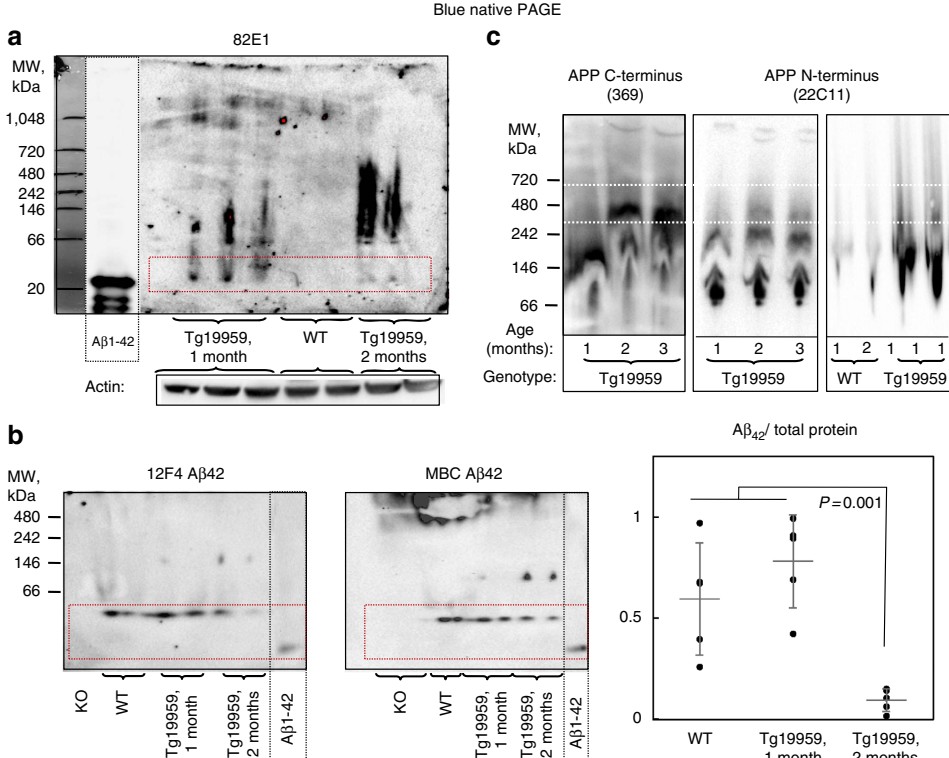

**Figure 4 | Native Aβ and APP complexes in Tg19959 mouse brains change with age.** (**a**) Blue native polyacrylamide gel electrophoresis (BN-PAGE) and subsequent Western blot of membrane-associated TBST fractions of mouse brain homogenates at 1 and 2 months of age. As detected by the human specific 82E1 antibody, human Aβ (dotted red box) in the Tg19959 mouse brain appears consistent with Aβ1-42 tetramers as well as a smear at higher molecular weights. Synthetic human Aβ1–42 was used as a molecular weight marker and positive control (dotted black box; underexposed). (**b**) BN-PAGE of membrane-associated fractions of mouse brain homogenates at 1 and 2 months of age and subsequent Western blotting with Aβ42 specific antibodies 12F4 and MBC42 detect the presence of low molecular weight Aβ42 bands (dotted red boxes). The low molecular weight Aβ42 band appears specific, since no band is observed in brain tissue homogenate from APP knockout mice. Statistical analysis of Aβ42 in brain homogenate: ANOVA ($P<0.01$) followed by Bonferroni's *post-hoc* comparisons test ($P<0.01$); grey lines indicate the mean ± s.d.; $N$ of 4 represents number of independent experiments. (**c**) Native APP in Tg19959 mouse brains changes with age. BN-PAGE and subsequent blotting with antibody 369 against the C-terminus of APP/CTFs and 22C11 against the N-terminus of APP show alterations in the molecular weight of APP with a 480 kDa band appearing at 2 months of age in Tg19959 mice (dotted white boxes), which is not seen in WT or 1 month old Tg19959 mice. Total protein concentrations were determined using the BCA assay and further controlled by a parallel SDS PAGE blotted with β-actin; protein loaded was 100 μg per well in all experiments. Blots are representative, $N=3$.

transgenic mice. Interestingly, with the increasing synaptic Aβ in Tg19959 mouse brains at two months of age, the overlap of Aβ42 with the presynaptic protein synaptophysin also became more evident. Moreover, at 3 months of age, when amyloid fibrils detected by antibody OC are even more evident in synapses of AD transgenic mouse brains, the pre- and post-synaptic proteins appear to overlap more; such an increased overlap of the pre- and post-synaptic markers suggests localized synaptic abnormalities (Supplementary Fig. 7). The presence of Aβ in synapses of wild-type mouse brains, and its early accumulation in synapses in transgenic mice, supports the conclusion that synapses are the sites where physiological Aβ starts to accumulate and aggregate, thereby mediating synapse dysfunction in pre-plaque AD brain. Taken together our results demonstrate alterations in the native states of Aβ and APP with pre-plaque β-aggregation, which is concomitant with synaptic alterations in a transgenic mouse model of β-amyloidosis.

## Discussion

Here we provide evidence that the physiological conformation of Aβ in brain on BN-PAGE is consistent with synthetic Aβ1-42 tetramers. Of note, tetramers were also the preferred state of freshly prepared Aβ1-42 run on the native gels (Fig. 4a;

Supplementary Fig. 5a). Native gels are less accurate when it comes to molecular weight, and running the same synthetic Aβ1-42 preparations on semi-denaturing and SDS–polyacrylamide gel electrophoresis (SDS–PAGE) gels showed progressive loss of the tetramer band, and indicated that this tetramer band runs closer to a 14 kD molecular weight marker (not shown). We also noticed that the Aβ42 specific antibodies detected the band consistent with Aβ42 tetramers in brain but not the synthetic Aβ1-42 tetramers (Fig. 4b). We previously reported that Aβ42 specific antibodies react predominantly to Aβ1-42 monomers[20]. However, the observations in Fig. 4b suggest that the structure of the Aβ band that appears consistent with a tetramer in brain differs from synthetic Aβ1-42 tetramers. It is possible that, for example, associated lipid components and/or N- and/or C-terminally modified Aβ peptides contribute to this difference between this low molecular weight Aβ complex in brain compared to synthetic Aβ42.

Importantly, we demonstrate that the loss of this physiological state of Aβ in AD transgenic mouse brain before plaque formation is concomitant with the appearance of increased β-sheet content by FTIR; however this is not ThS or OC antibody positive, pointing to a non-fibrillar character of early β-sheet formation. Although, the differences between transgenic and wild type animals in the μFTIR spectra related to β-structures and

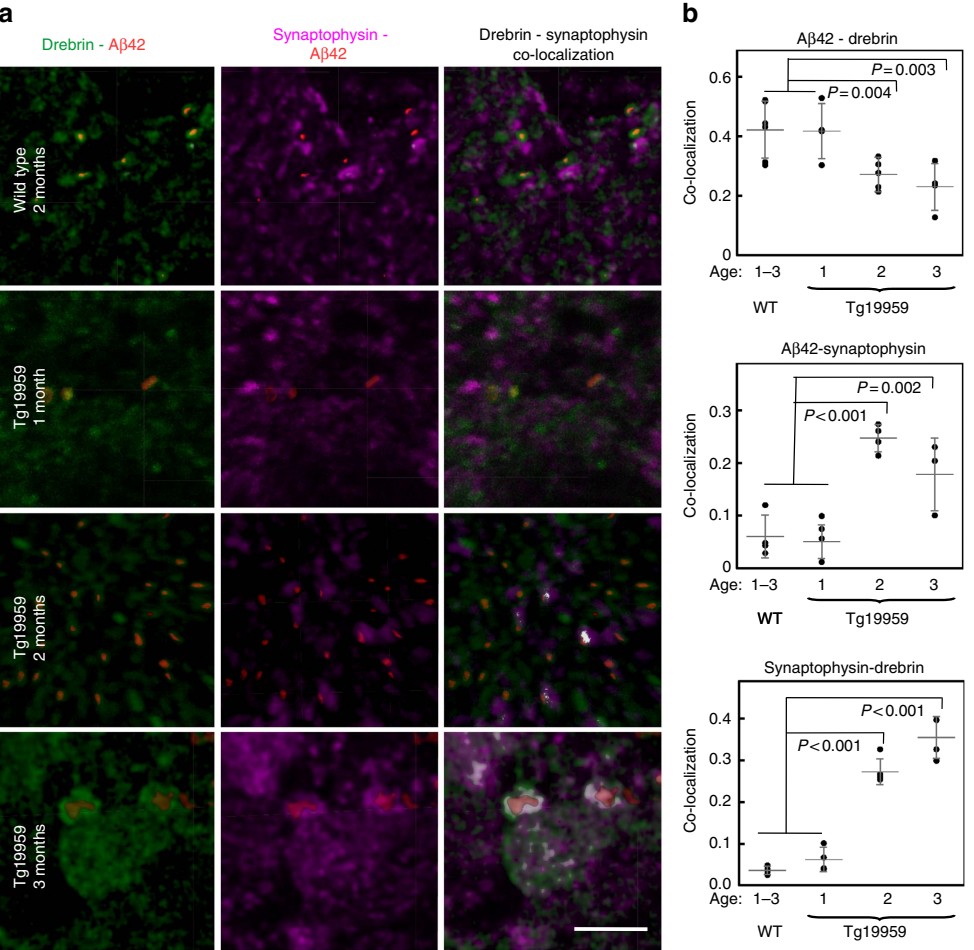

**Figure 5 | Aβ42 disrupts the cytoarchitecture of dendritic terminals in Tg19959 mouse brains with age.** (**a**) High-resolution confocal microscopy images of immunofluorescently labelled Aβ42 (red), the post-synaptic protein drebrin (green) and the pre-synaptic protein synaptophysin (magenta) in 2-month-old wild type, and 1–3 months-old Tg19959 mouse brain cortex. Iso-surface rendering of Aβ42 more clearly reveals Aβ42 within drebrin positive compartments. Note the presence of Aβ42 also in wild-type mice and the increasing levels of Aβ42 with age in Tg19959 mice. In the far left panel, the white colour shows co-localization of drebrin and synaptophysin, which is evident in 2 and 3 month-old Tg19959 mice. Scale bar is 2 μm. For an overview, low magnification images are shown in Supplementary Fig. 8. (**b**) Statistical analysis of the co-localization of Aβ42 with pre- and post- synaptic markers, as well as for the overlap of synaptophysin with drebrin: ANOVA (P < 0.01) followed by Bonferroni's post-hoc comparisons test (P < 0.01); 1 is equal to 100% of co-localization, grey lines indicate the mean ± s.d., N = 4.

age-related increases in Aβ42 are evident, development of new antibodies which are both specific to β-structures and to Aβ will be helpful to determine whether the β-sheet structures detected by μFTIR in brain tissue of transgenic animals originate from Aβ42. Further, we present evidence consistent with aggregation of APP concomitant with Aβ pathogenesis, which may be contributing to the pre-plaque increase in β-structure. Since the increase of Aβ is concomitant with an apparent change in APP structure, and since *in vitro* experiments support that Aβ may bind APP, it is possible that binding of Aβ may cause changes in APP structure that interfere with physiological processing of APP. Dystrophic neurites, associated with amyloid plaques, are known to accumulate APP, as well as Aβ42 and the β-site amyloid precursor protein cleaving enzyme 1, supporting the concept that such dystrophies are the nidus of amyloid plaque formation[20,25].

It is of considerable interest that the destabilization and loss of the native state of normally tetrameric transthyretin has been found to promote subsequent aggregation of transthyretin into the β-sheet rich pathologic amyloid found in patients with transthyretin amyloidosis. Moreover, pharmacological stabilization of the physiological transthyretin tetramers led to the approval of the first drug that slows down the progression of an amyloidosis in the clinic[26–28]. Growing evidence also suggests that the native oligomeric structure of α-synuclein associated with Parkinson's disease is a tetramer[29]. Thus, the current study supports the concept that the native and physiological conformations of proteins linked pathologically and genetically to AD might also be altered in the disease. Stabilization of the physiological structure might therefore be considered among novel therapeutic approaches.

## Methods

**Animals.** All mouse experiments were compliant with the requirements of the Ethical Committee of Lund University. Experimental AD transgenic mouse group: female Tg19959 mice, which harbour Swedish and Indiana familial AD mutations (KM670/671NL and V717F) in human APP under the control of the hamster PrP promoter[12]. Control group: female C57/B6SJL wild-type mice. Primary neuronal cultures were derived from cerebral cortex and hippocampus of Tg (hAPPswe, PSEN1dE9)85Dbo/Mmjax (APP/PS1; Jackson Labs) or wild-type mice at embryonic day 15 as described[30]. All mice were screened for the presence of the human APP695 transgene by PCR.

*Experimental design.* Brain material (sections or homogenates) of female Tg19959 and wild-type mice at 1, 2 and 3 month of age, N of 3–4 animals per genotype/age, were collected for the study. Brain sections were used for μFTIR and immunofluorescence imaging; brain homogenates (cortex) were used for biochemical studies.

*Experimental procedures*. Female Tg19959 and wild-type mice were killed at 1, 2 and 3 month of age. Mice were deeply anaesthetized with 100 mg kg$^{-1}$ 1:10 ketamine:xylazine administered by intraperitoneal injection. For μFTIR and biochemical analysis, animals were first perfused transcardially with phosphate-buffered saline (PBS), and then brains were removed from skulls and deeply frozen in liquid nitrogen. For immunohistochemistry, brains were fixed by transcardial perfusion with PBS followed by 4% paraformaldehyde (PFA) in 0.1 M PBS (pH 7.4) at room temperature (RT) and post-fixed by immersion in 4% PFA in 0.1 M PBS at 4 °C overnight.

**Aβ1-40 and Aβ1-42 peptides.** Synthetic Aβ1-40 (DAEFRHDSGYEVHHQKLVF FAEDVGSNKGAIIGLMVGGVV) was purchased from JPT (Germany). Stock solution of 500 μM synthetic Aβ1-40 (JPT) was prepared in 10 mM HEPES (Sigma-Aldrich) with 0.02% NH$_3$ at pH 11 (pH of the non-aggregated peptide stock); aliquots were kept at −80 °C until use. Aβ1-40 oligomers were prepared in the presence of ions of Cu(II) as described[31], with a minor change: Cu$_2$SO$_4$ was added in a stoichiometric ratio to Aβ1-40 stock solution, and then pH was changed to pH 7.4 to allow Aβ aggregation.

Small angle x-ray scattering (SAXS) analysis of the Aβ1-40 Cu(II)-induced oligomers was done at the beamline X33 EMBL DESY synchrotron (Germany). SAXS data was acquired for 1 mg ml$^{-1}$ Aβ1-40-Cu(II)O in 10 mM Hepes pH 7.4 after 24 h of incubation at 37 °C before the experiment. The fit for the model (spheres) was calculated using GNOM (ref. 32). *De novo* three-dimensional reconstruction of the scattering entity of Aβ1-40-Cu(II)-induced oligomers was performed using DAMMIN (ref. 32) after 10 independent DAMMIN reconstructions.

Synthetic Aβ1-42 (DAEFRHDSGYEVHHQKLVFFAEDVGSNKGAIIGLMVG GVVIA) was purchased from Tocris Bioscience (USA). The peptides were dissolved in cold DMSO (Sigma–Aldrich) at a concentration of 250 μM, divided into aliquots of 50 μl and kept at at −80 °C until use. Recombinant Aβ1-42 (M-DAEFRHDSGYEVHHQKLVFFAEDVGSNKGAIIGLMVGGVVIA), 20 μM in 20 mM phosphate buffer (pH 7.4), was a gift of Prof. Sara Linse (Lund University). Recombinant Aβ1-42 was prepared and purified as described[33], and kept on ice until use (1–2 h).

Fibril growth assays were initiated by placing the 96-well plate at 37 °C under quiescent conditions. The ThT fluorescence was measured through the bottom of the plate every 60 s with a 440 nm excitation filter and a 480 nm emission filter in a plate reader (Fluostar Omega BMG Labtech). To monitor fibril formation, Thioflavin T (ThT) kinetic assays were used. For ThT kinetics, 20 μM Aβ1-42 was incubated in the presence 6 μM ThT at 37 °C without agitation in a 96-well plate of black polystyrene with a clear bottom and PEG coating (Corning 3881). Since ThT fluorescence intensity at 480 nm is proportional to the mass of amyloid fibrils, the kinetic evaluation of the aggregation reactions showed that after several hours of incubation, Aβ1-42 formed β-sheet fibrils. For μFTIR measurements, in parallel to the ThT kinetic assays, Aβ1-42 was incubated without adding ThT in low-bind Eppendorf tubes at 37 °C under quiescent conditions.

**Synchrotron-based μFTIR.** Approximately 20 μm thick coronal brain cryosections were mounted onto clean 1 × 1 mm$^2$ CaF$_2$ windows (Crystran Ltd.); to avoid protein degradation sections were stored at −80 °C until measurements as described[2]. Sections were cryo-dried before measurements. Primary neurons were seeded directly on CaF$_2$ spectrophotometric windows and grown for 12 or 19 days. Cultures were washed with PBS, fixed for 20 min with 4% PFA in PBS, washed with 20 mM phosphate buffer (PB) and stored at −80 °C until measurements. Samples with neurons were dried before measurements. Samples of Aβ1-42 monomers and fibrils were prepared in the same way as the μFTIR samples from brain tissues. Approximately 5 μL drops of monomeric and fibrillar Aβ1-42 fractions were placed on CaF$_2$ spectrophotometric windows, snap frozen and cryo-dried before measurements were taken.

μFTIR spectroscopy was performed at beamline D7, MAX-IV Laboratory, Lund University, Sweden as described[34,35]. μFTIR spectra were collected from mouse brain sections of cortex and CA1 hippocampus (as shown in Supplementary Fig. 2a). The instrument set-up combines a Hyperion 3,000 microscope and a Bruker IFS66/v FTIR spectrometer. First, μFTIR maps were recorded in off-line mode using a conventional thermal light source for 'overview mapping' of larger tissue areas (4–6 mm$^2$) using 50 μm aperture diameter. Since the brightness of conventional thermal infrared sources is inherently limited near the diffraction limit, a synchrotron infrared source, which is about 100–1,000 times brighter than a conventional thermal source, was used. The high flux density of the synchrotron source allows smaller regions to be probed with an acceptable S/N ratio[4]. FTIR spectra were collected from cultured neurons at MAX-IV Laboratory and the SOLEIL synchrotron (Gif-sur-Yvette, France) using the SMIS beamline (Supplementary Fig. 2d). The measuring range was 900 − 4,000 cm$^{-1}$ and the spectra collection was done in transmission mode at 4 cm$^{-1}$ resolution, 8 × 8 μm$^2$ aperture dimensions, from 500 to 1,000 co-added scans. Background spectra were collected from a clean area of the same CaF$_2$ window.

**FTIR spectral analysis.** Analysis of FTIR spectra was performed using the OPUS software (Bruker). After atmospheric compensation, spectra exhibiting strong Mie

scattering were eliminated. For all spectra, a linear baseline correction was applied from 1,200 to 2,000 cm$^{-1}$. After background subtraction and vector normalization, derivation of the spectra to the second order was used to increase the number of discriminative features to eliminate the baseline contribution. Derivation of the spectra was achieved using a Savitsky − Golay algorithm with a nine-point filter and a polynomial order of two. The β-aggregation level of proteins was studied by calculating the peak intensity ratio between 1,620 and 1,640 cm$^{-1}$, corresponding to β-sheet structures and the maximum corresponding mainly to α-helical content at 1,656 cm$^{-1}$. An increase in the 1,620–1,640 cm$^{-1}$ component is considered a signature of amyloid fibrils[10,11,14,36]. We were not able to detect changes in the range of 1,690 cm$^{-1}$ that were previously described to differentiate antiparallel and parallel Aβ in a study on purified Aβ using FTIR (ref. 37), likely due to the greater complexity of the tissue samples.

**Immunohistochemistry.** PFA fixed mouse brains were cut into 40 μm thick sections on a Leica SM 2010R freezing microtome. Sections were kept in storage buffer composed of 30% sucrose and 30% ethylene glycol in PBS at −20 °C until use. Dual and triple immunolabelling combined with ThS of free-floating sections were done as described in ref. 38 using optimal working dilutions recommended by the manufacturer. Amyloid fibrils were visualized with rabbit polyclonal antibody OC (Merck Millipore, AB2286); the free Aβ42 C-terminus was recognized by antibody 12F4 (BioLegend, Covance SIG-39142, 1/1,000); antibody 22C11 (Merck Millipore, MAB 348) was used for the human/mouse APP N-terminus. For pre-synaptic and post-synaptic labelling mouse monoclonal synaptophysin antibody (Merck Millipore, MAB5258) and rabbit polyclonal drebrin antibody (Abcam, AB11068) were used respectively. All primary antibodies are summarized in Supplementary Table 1. For dual and triple label ThS staining, sections were incubated in 0.001% ThS in 70% ethanol for 20 min, and then rinsed sequentially with 70, 95 and 100% ethanol after immunofluorescent labelling as described[38].

**Confocal microscopy.** Images were obtained using a Leica TCS SP8 confocal microscope (Leica Microsystems) equipped with Diode 405/405 nm and Argon (405, 488, 552 and 638 nm) lasers with an HP PL APO 63x/NA1.2 water immersion objective. Autoquant (MediaCybernetics) was used for image deconvolution. Two-dimensional images obtained by confocal microscopy were quantified and reconstructed into 3D volumetric data sets using Imaris (Bitplane). Semi-opaque iso-surfaces were defined individually for each channel from the deconvolved data and rendered as semi-opaque, solid surfaces.

**Biochemical analysis of mouse brain tissue.** Forebrains were homogenized in five volumes of buffer composed of 20 mM Tris–HCl, 50 mM NaCl pH 7.8 and HaltTM protease inhibitor cocktail (Thermo Fisher Scientific). After 30 min incubation on ice, a first fraction was collected as a TBS dispersible fraction. A TBS-T membrane-bound fraction was collected after the TBS insoluble pellet was re-suspended in 1% Triton, 20 mM Tris–HCl, 50 mM NaCl at pH 7.4, incubated 30 min on ice and centrifuged. To avoid artificial protein aggregation or segregation, brain homogenates were centrifuged at low speed, at 10,000 g, at 4 °C for 30 min as described[24]. Protein amounts were determined using BCA protein assay (ThermoFisher Scientific). The protein load was also controlled by a parallel SDS–PAGE blotted with β-actin (Sigma A5316, 1/5,000). Samples were used immediately for BN- or SDS–PAGE and co-immunoprecipitation assays. For BN-PAGE, only freshly prepared samples of 100 μg total protein in 4 × Native PAGE sample buffer (Invitrogen) were electrophoretically resolved in a precast native 4–16% Bis-Tris gel (Invitrogen) according to the manufacturer's protocol. Native-Mark unstained protein standards (Invitrogen) were used as molecular weight markers. Before protein transfer, the gels were washed with running buffer containing 1% SDS for 20 min. The protein load was also controlled by a parallel SDS–PAGE blotted with β-actin (Sigma A5316, 1/5,000). For SDS–PAGE, brain homogenates (50 μg of total protein) were electrophoretically resolved in a precast NuPAGE 4–12% Bis-Tris gel system (Invitrogen). Proteins were transferred from BN and SDS gels onto polyvinylidene difluoride (PVDF) membranes (iBlotR, NovexR, Life Technologies). Unstained molecular weight markers were visualized using Ponceau S (SigmaAldrich) and digitally marked due to faint bands on the blue membranes. Membranes were boiled in PBS, pH 7.4 in a microwave oven for 5 min, and washed in PBS-TweenTM (Medicago). To block unspecific binding, membranes were incubated in 5% non-fat dry milk (Sigma) diluted in PBS-Tween for 1 h at RT. For immunodetection, membranes were incubated for 24 h at 4 °C with primary antibodies against Aβ/APP: 6E10 and 4G8 (BioLegend, Covance #SIG-39300 and #SIG-39220 respectively, 1/1,000), human/mouse APP N-terminus: 22C11 (Merck Millipore, MAB 348, 1/1,000), human APP: P2-1 (Thermo Fisher Scientific, 1/1,000), and human/mouse APP C-terminus antibody 369 (Supplementary Table, 1/1,000). The Aβ N-terminus was recognized by Aβ/APP monoclonal antibody 82E1 (IBL International #18582, 1/1,000), and the Aβ42 C-terminus was recognized by Aβ42 end-specific antibody from Life Technologies (#700254, 1/1,000) and 12F4 (BioLegend, Covance, SIG-39142, 1/1,000); MBCAβ42 (1/1,000); Chemicon (#AB5078, 1/1,000); primary antibodies are summarized in Supplementary Table 1. Blots were developed with enhanced chemiluminescence (SuperSignalR West Femto) detection system (Thermo Fisher Scientific). Synthetic Aβ1-42 (Tocris Bioscience) was used as positive and/or

negative control. All BN-PAGE blots were developed with standard chemiluminescence exposure times of 2–5 min, up to maximum exposure times of 1 h to detect even minor amounts of Aβ. All western blots were performed at least three times; only for the confirmation of the Aβ band in Fig. 3b, three additional Aβ antibodies were used to confirm the 3 blots done with antibody 82E1 and to show that this Aβ includes Aβ42. For quantification, 4 gels blotted with Aβ42 antibodies were pooled together.

**Statistical analysis.** One-way ANOVA followed by Bonferroni's *post-hoc* comparison tests were performed in all statistical analysis. Since no significant differences were found in the wild-type brains, the data from all the wild types were pooled and used in the statistical calculations. Statistics with a value of $P < 0.01$ were considered significant.

**Data availability.** The data that support the findings of this study are available from the corresponding authors on reasonable request.

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

## Acknowledgements

We thank Vladimir Denisov and Sara Linse for critical input to this manuscript. We thank Per Persson for the access to FTIR equipment. This work was funded by grants from Alzheimerfonden, Interreg Öresund-Kattegat-Skagerrak (EU), the Swedish Research Council, MultiPark and the Segerfalk, Anna-Lisa Rosenberg and Åhléns foundations.

## Author contributions

O.K., G.K.G. designed experiments, analysed data and wrote the manuscript. G.K.G. supervised the research. O.K., K.W., I.M. and B.I. performed experiments. A.E. and P.U. assisted in μFTIR data acquisition and discussed the FTIR results. J.C. supervised the experiments described in Figure 2. All authors discussed the results and contributed to the scientific discussion.

## Additional information

**Competing financial interests:** The authors declare no competing financial interests.

