## [Peer Review File · Nature Communications]

Reviewers' comments:

Reviewer #1 (Remarks to the Author):

The manuscript by Klementieva et al, describes a novel experimental approach to structurally characterise the differences between Alzheimer's transgenic mice (APP^{Swe/ind}) and controls at different ages. The work is very interesting. They then go further to purify assemblies and characterise these using native page and western blotting to identify different assemblies of Abeta. Finally, Ab is localised to synaptic terminals. The approach and the work are novel and of interest to the field.

Overall the paper is well written although I think that the text could be made clearer and more concise.

I do have some concerns however:

Although I can see that there are some differences between tg animals and non-tg animals in the uFTIR and that this changes with age in tg Animals, I don't think we can be absolutely sure that this arises from Abeta and not some other changes in the brain tissue. The antibody used recognises the C terminus so I assume picks up all conformations. It would be preferable to use an antibody that preferentially picks up oligomers or fibres, this would support the FTIR better. In fact, general use of oligomer specific antibodies could add to the work.

The neuron culture work was done on embryonic cells, and I am not sure that this is strictly comparable to the other work where the tissue is from much older animals. Is there any evidence of localisation to synaptic terminals in brain tissue from the older animals?

The native page approach is interesting, but the results shown in the figures are a little hard to be convinced by. I realise that this is an issue of using native page. I would like to see the full data rather than just "windows" showing the results and I would like to see that a secondary only control has been performed to avoid any potential cross reactivity (which is known to be a very significant issue with Abeta antibodies and western blotting)

Minor comments

Some of the spelling and sentences need editing and check (for example : monocular instead of molecular and AD instead of Abeta, both pg 6).

In the abstract and in the discussion, the authors mention stabilising precursor or native structures. I see why this is introduced, but I am not sure that this really relevant to the current paper and it is clear that a lot more work will need to be done to provide clear evidence for the conclusions regarding b-sheet "tetramers".

Reviewer #2 (Remarks to the Author):

A. Summary of the key results

In this work the authors characterized conformation change of A β 42 before the amyloid plaque deposition in an AD Transgenic mouse model using a combination of Synchrotron-based μ FTIR imaging, BN-PAGE and conformational specific antibodies. Their primary finding is that under physiological condition, A β 42 initially exists as 20 kDa (~4mer) protein complex in the brain and forms non-fibrillar aggregates before transforming in structure to form amyloid plaques. A secondary finding is that 20 kDa A β appears to be localized to neurons. The paper further showed the interactions between the A β 42 aggregates and the APP.

Evidence that is supportive of the paper's conclusions:

1. Increasing FTIR ratio of (β -sheet absorbance)/(α -helix absorbance) was observed with aging of the mice, indicating conversion of structure aggregates are rich in β -structure.
2. The pre-plaque aggregates are not positive for assay with fibril sensitive fluorescent dye (ThS) and antibody (OC).
3. The early-age mouse model showed \sim 4mer A β 42 bands in the (non-denaturing) BN-PAGE bands were became weaker with aging for the mouse model, corresponding to deposition of amyloid plaques. This correlation was interpreted to indicate the transition from the conformation of A β 42 aggregates.

B. Originality and interest

The idea that there exists a tetrameric state of A β in vivo that precedes plaque is novel to the best of my knowledge. I am skeptical that this state would be a good drug target, since its toxicity is not known. However, the evidence that this state may exist makes me feel that this paper is of great interest.

C. Data & Methodology

Overall, the work reports novel results and seems to be reasonably well done, but there are some points of confusion that should be addressed. These points could probably be addressed through revision of the text without requiring new experimental data (Please see part F).

D. Appropriate use of statistics and treatment of uncertainties.

The authors used ANOVA followed by Bonferroni's post-hoc comparison test.

Could the authors comment on the reproducibility of the FTIR spectra? The authors may wish to consult a similar in situ FTIR study of protein aggregation in human tissues, (D. Ami, F. Lavatelli, et.al. Scientific Reports 6, 29096 (2016)).

E. Conclusions: robustness

Based on the experimental observation that A β 42 forms tetramer protein complex under physiological condition and then proceed to toxic β -structure oligomers and finally plaques, the authors claimed that stabilizing the tetramer might slow down the progression of the amyloidosis and became a potential therapeutic strategy. However, I am still skeptical that this state would be a good drug target, since its toxicity is not known and there are many group have shown showing evidence that low molecular weight A β aggregates are neuro toxic.

F. Suggested improvements: experiments, data for possible revision

Points of confusion:

1. The basis for interpretation of the different molecular structural states of A β in the in vivo μ FTIR images is shown in Supplementary Figure 5. The authors should consider a more prominent position for Supplementary Figure 5 (e.g., in the main text). They should also make it more clear that interpretations are based on FTIR analysis of spectra from the "model compounds" that do not necessary match structures that would be found in vivo. The interpretation of peak position would be summarized as follows:
 - a. FTIR measurement of synthetic unstructured A β 40 monomer indicates a peak just above 1640 cm^{-1} . Since A β 40 and A β 42 monomers are generally believed to be unstructured in solution, this is probably a reasonable interpretation. There are some claims in the literature about α -helical content for A β monomers, but alpha-helical structures seem to yield similar peak positions in FTIR. This signature was not observed in vivo, suggesting that no unstructured monomers were present.

b. The interpretation of the "oligomer" and "fibril" FTIR signals seems less conclusive. It may be unreasonable to expect perfectly conclusive results from in vivo studies like this one, but the paper should at least make it clear what assumptions and conjectures are being made. FTIR data on a synthetic Cu-induced A β 40 oligomer exhibits an FTIR peak at 1630 cm⁻¹ (Supplementary Figure 5). It is then assumed that any signal at 1630 cm⁻¹ for the in vivo measurements indicate the presence of an oligomer. FTIR measurement of A β (40) fibrils produced a peak at ~1623 cm⁻¹, leading to the interpretation that signals at this peak position indicate fibrils. However, it is not clear that the position of a single FTIR peak is sufficient to definitively identify oligomer or fibril structure. For example, data in (E. Cerf, R. Sarroukh, et. Al. *Biochem. J.* 421, 415-423 (2009)), a widely respected paper on FTIR of A β 42 aggregates, indicate that the main FTIR signal from oligomer and fibril β -sheets both show up at 1630 cm⁻¹. It is known that A β 40 and A β 42 show very different aggregation characteristics (especially in terms of oligomers) and it is not clear that the in vitro-generated Cu²⁺ A β 40 oligomers indicate anything about oligomers that may form in vivo. Furthermore, the BN-PAGE data in Figure 3 indicate that aggregates in the 100-400 kDa range co-exist with tetramers in vivo. What FTIR signatures would one expect from larger non-fibrillar aggregates? In the end, the μ FTIR spectroscopic imaging data seem to indicate that A β structural changes occur in vivo, but it is not obvious how to interpret these changes. Note that the BN-PAGE data support the interpretation that the early aggregates are not fibrils.

2. Supplementary Fig.1 is meant to show that the process of freezing and drying does not induce fibril formation (or any other aggregation). FTIR samples show distinct spectral patterns, specifically, the absorbance peak indicating β -sheet structure only occurs in fibrils but not monomers. It is problematic that the statement about the sample preparation is hidden in the middle of the caption, making the figure capture hard to follow [..."frozen and dried (in the same way as the brain tissue slides for μ FTIR imaging)..."]. In addition, it should be emphasized that lack of aggregation in the in vitro preparation (A β 42 fibrils and monomers) does not necessarily indicate that aggregation would not have been affected in the tissue samples. The in vitro and in vivo environments are different and the in vivo environment contains both A β 40 and A β 42 (see the first author's own paper, *Biomacromolecules* 2013). While it may be impossible to provide an ideal control experiment for the in vivo measurement, this limitation should at least be acknowledged.

3. In Kelementeva's (the first author) *Biomacromolecules* paper, she used the same AD Tg mice and study the A β aggregation, it seems the A β 42/A β 40 ratio was increased in the 5 month old Tg mice and there is still a strong 20kDa band shown in the non-SDS PAGE, this is not consistent with the current paper, which claims the 20kDa A β 42 protein complex will be eliminated as the Tg mice age. Could this apparent contradiction be explained?

4. The authors conducted experiments to show co-localization of A β 42 and APP. This would appear to be a notable result. However, little is offered in the paper regarding interpretation of this finding. Why is it significant that A β interacts with APP?

G. Reference: appropriate credit to previous work?

The references seem to be appropriate.

H. Clarity and context: lucidity of abstract/summary, appropriateness of abstract, introduction and conclusions

Most of the text is reasonably clear. The sentence "Thus, the current study supports the novel concept that also in AD destabilization of native conformations of proteins linked pathologically and genetically to the disease might be critical for this most common neurodegenerative disease, and that AD may, therefore, be similarly amenable to treatment by stabilization of normal protein complexes." on page 8 is too long and not clear.

Reviewer #3 (Remarks to the Author):

NCOMMS-16-12737

Klementieva, et al.,

Pre-plaque conformational changes in Alzheimer's disease-linked A β /APP

A. Summary: The authors report that Abeta and APP containing beta sheet deposits in transgenic mice accumulate prior to the time that they become thioflavin and OC positive, which are conformational probes of different types of amyloid structure. The authors show that cultured neurons from Tg mice accumulate similar beta sheet deposits with increasing incubation time. They also show that Abeta and APP aggregates accumulate concomitant with the increase in beta sheet content and that exogenously added Abeta42 co-aggregates with APP in these neurons. Finally, the authors show that The Abeta42 that accumulates is associated with synapses and that the cyto-architecture of the synapses is altered in the vicinity of plaques. The conclusion is that Abeta and APP misfold, co-aggregate and adopt beta sheet conformation before they form OC and ThS positive aggregates.

B. Originality and Interest.

The results are generally original and interesting. It is known that the cyto-architecture of the neuropil is altered upon amyloid deposition but not specifically early before accumulation of OC+ and ThS+ aggregates.

C. Data

1. P4, paragraph1: "prior to amyloid plaque formation as determined by thioflavin S (ThS) staining". Thioflavin S staining is not a reliable definition of a plaque even though it has been used as an operational definition for a number of years. There is increasing evidence that it is just another probe of amyloid conformational differences. Some fibrils bind it; other fibrils don't. Some oligomers bind it; others don't. There are many published examples of this, but the most recent one that sticks in my mind was published this past week on the Osaka mutation Abeta fibrils by Matthew Elkins and Mei Hong from MIT in JACS. Similarly, it is known that OC reacts with beta sheet aggregates that are oligomeric and not morphologically fibrillar. This does not challenge the author's observation of conformational differences, but only the claim that deposits that are not ThS positive are not fibrils or plaques or that OC positive deposits are necessarily fibrillar.

D. Statistics

No problems noted.

E. Conclusions

The conclusions are reasonable and appropriate and are supported by the data.

F. Suggested improvements.

1. Vincent Raussens has shown that parallel and antiparallel beta sheet amyloids can be distinguished by their 1692.1639 ratio of their FTIR spectra and he has speculated that this may form the difference between antiparallel, oligomeric forms that react with conformation dependent antisera, A11 and parallel, fibrillar forms that react with OC sera (Biochem J. 2009 Jul 15;421(3):415-23. doi: 10.1042/BJ20090379.). Can you interpret your FTIR spectra in terms of parallel and antiparallel beta sheet content? Do the 2 month deposits that do not stain with OC, stain with A11?

Minor points:

1. Introduction: It isn't necessary to demean other approaches to validate the approaches used here and this verbiage may not endear the authors to practitioners of alternative approaches who may question the "non-destructive" nature of synchrotron IR radiation.

2. "Similar high monocular weight A β was also observed on BN-PAGE in aged APP23 AD transgenic" Molecular weight

Reviewer 1

We were happy that the reviewer found our manuscript “*very interesting*” and noted “*The approach and the work are novel and of interest to the field.*”

The reviewer made specific suggestions of making the text clearer and more concise, revising the figure with native PAGE, and adding control experiments to test possible cross-reactivity of antibodies.

Point 1. *Although I can see that there are some differences between tg animals and non-tg animals in the μ FTIR and that this changes with age in tg Animals, I don't think we can be absolutely sure that this arises from Abeta and not some other changes in the brain tissue.*

Response: We agree with the reviewer and have modified our Discussion as follows (page 8, lines 206-210):

“Although the differences between transgenic and wild type animals in the μ FTIR spectra related to β -structures and age-related increases in A β 42 are evident, development of new antibodies which are both specific to β -structures and to A β is required to determine whether the β -sheet structures detected by μ FTIR in brain tissue of transgenic animals originate from A β 42.”

Point 2. *The antibody used recognises the C terminus so I assume picks up all conformations. It would be preferable to use an antibody that preferentially picks up oligomers or fibres, this would support the FTIR better. In fact, general use of oligomer specific antibodies could add to the work.*

Response: Several different antibodies and dye (Thioflavin S) were used, including C-terminus specific A β antibodies and the conformation specific antibody OC. In previous work we have seen that C-terminal 42 antibodies preferably detect monomeric Abeta42 (Takahashi RH et al., 2004), which might be expected, since the C-terminus is highly aggregation prone. In contrast, antibody OC detects fibrillar structures, although as reviewer 3 points out there are different forms of fibrils that can be differentially recognized. In response to the reviewer’s point we have now rewritten this sentence in the original manuscript: “*...adjacent Tg19959 brain sections to those showing early β -aggregation by FTIR did not show evidence of fibrillar A β using the conformation-specific antibody OC⁹ or ThS, both indicators of fibrillar protein aggregates.*”

The sentence is now changed to (page 4-5, lines 104–110):

“To complement the μ FTIR data, we used the conformation-specific antibody OC¹⁵ and ThS, which are both considered indicators of fibrillar amyloid structures. However, the adjacent Tg19959 brain sections to those showing early β -sheet structures by FTIR did not show evidence of amyloid fibrils with antibody OC or ThS (Supplementary Figs. 3 and 4). The absence of ThS and OC antibody labelling may be due to the low concentration of fibrils and/or dye specificity, that thioflavin S and OC antibody labelling do not detect all types of fibrils^{16,17}, or due to a non-fibrillar nature of the β -sheet structures.”

We note that since receiving the reviews, we again tried oligomer-specific antibody A11, which had not worked for us in the past, and we again could not see convincing labelling. In addition, we found a reference from the lab of Charles Glabe indicating that A11 is not effective for mouse brains (Kayed R et al., 2010). Specifically, Kayed R et al., 2010, wrote: “*We also investigated whether the monoclonal IgGs can detect the accumulation of PFOs in human AD brain and Tg2576 and 3xTg-AD transgenic mouse brains. No specific staining of plaque deposits was observed by immunohistochemistry in human AD brain (Figure 6) or transgenic mouse brain (data not shown), indicating that the Mabs do not stain plaques, consistent with their lack of reactivity with amyloid fibrils. It is not yet clear whether the oligomers recognized by these antibodies are absent from human AD or Tg mouse brain tissue. The antibodies exhibit low background reactivity on tissue, indicating that they do not detectably cross react with normal proteins. Further*

experiments with higher resolution and sensitivity, such as immuno electron microscopy and immunoprecipitation will be necessary to determine whether these oligomers are detectable in vivo...)."

We also were not able to detect specific labelling of A11 on dot blots of the Tg compared to wild type mouse brain tissue (not shown).

Point 3. *The neuron culture work was done on embryonic cells, and I am not sure that this is strictly comparable to the other work where the tissue is from much older animals.*

Response: The reviewer raises a legitimate concern about the use of embryonic neurons when studying an age-related disease. However, primary neurons have been shown to undergo accelerated ageing in vitro. Work by many groups have shown accelerated age-related features in primary embryonic neurons grown in culture, which unlike neurons *in vivo*, die after about 4 weeks in culture; for example, the lab of C. Dotti showed that the lipid composition of neurons changes with time in culture similar to what occurs in ageing brain *in vivo* (e.g. Martin MG et al., 2008). Primary neurons also show age-related rises in markers of oxidative stress, as well as accumulation of A β and AD-like alterations in synapses with time in culture in AD transgenic neurons that mirror what is seen in brains of AD transgenic mice with age as well as in human AD brains (reviewed in Gouras GK et al. 2010). However, since this certainly is a valid point, we now changed our former sentence on page 6:

"Taken together, these results support the conclusion that A β begins to form β -sheet structures prior to amyloid plaque formation."

to (page 5, lines 127–129):

"Although age of neurons in culture²¹ is not directly comparable to age of neurons in brain, these results support the conclusion that β -sheet structures can be formed within neurons."

Point 4: *Is there any evidence of localisation to synaptic terminals in brain tissue from the older animals?*

Response: Yes, prior evidence both in older AD transgenic mice and also in human AD brain shows A β localization to synaptic terminals (Takahashi RH et al., 2002; Gylys KH et al., 2004; 2007). However, we note that the current manuscript is focused on the early stage of A β aggregation, which has not been examined previously.

Point 5. *The native page approach is interesting, but the results shown in the figures are a little hard to be convinced by. I realise that this an issue of using native page. I would like to see the full data rather than just "windows" showing the results and I would like to see that a secondary only control has been performed to avoid any potential cross-reactivity (which is known to be a very significant issue with Abeta antibodies and western blotting)*

Response: Based on this comment new blots have now been added; see revised Figure 3 and its associated figure legend. In addition, the secondary antibody control experiments were now done and the Western blots included in revised Supplementary Figure 6.

Point 6: *Some of the spelling and sentences need editing and check (for example: monocular instead of molecular and AD instead of Abeta, both pg 6).*

Response: We thank the reviewer for pointing these out and have now corrected these errors and carefully edited our manuscript.

Point 7: *In the abstract and in the discussion, the authors mention stabilising precursor or native structures. I see why this is introduced, but I am not sure that this really relevant to the current paper and it is clear that a lot more work will need to be done to provide clear evidence for the conclusions regarding b-sheet "tetramers".*

Response: We fully agree that more work needs to be done to understand the nature of the low molecular weight A β complexes. We have now deleted the end of the following last sentence of our original abstract to reduce our previous emphasis on this issue (page 2, lines 37–38):

“These pre-plaque changes in the states of A β and APP suggest novel approaches for AD therapy based on stabilisation of protein complexes in their physiological states.”

We also reduced our emphasis on this more speculative aspect in our revised Discussion.

Reviewer 2

We appreciate the reviewer’s careful and extensive comments, as well as interest in our work expressed as e.g.: *“Overall, the work reports novel results and seems to be reasonably well done, but there are some points of confusion that should be addressed. These points could probably be addressed through revision of the text without requiring new experimental data”*.

Specifically, reviewer 2 suggested to clarify the interpretation of the FTIR peak positions, to acknowledge the limitations of the *in vitro* experiments used to test the sample preparation for FTIR to support that β -sheet structures may originate from non-fibrillar species, and to revise our discussion on potential A β 42 and APP interaction.

Point 1: *Could the authors comment on the reproducibility of the FTIR spectra?*

Response: To demonstrate the reproducibility of the FTIR spectra we now also plotted the individual spectra from different animals of the same genotype and added these plots to Supplementary Figure 2. Next to adding the FTIR spectra of different mice in Supplementary Fig. 2, we also added the following sentence in the legend for Figure 1 on page 19, lines 472–473:

“FTIR spectra from different animals of the same age and genotype are shown in Supplementary Fig. 2c.”

Point 2: *The authors may wish to consult a similar in situ FTIR study of protein aggregation in human tissues, (D. Ami, F. Lavatelli, et.al. Scientific Reports 6, 29096 (2016)).*

Response: We appreciate that the reviewer pointed out this paper by Ami et al., 2016. We both revised our text and now cite the paper in the following new sentence (page 4, lines 88–97):

“To better resolve the peak positions for β -sheet structures around 1627 cm^{-1} and the band centred at 1656 cm^{-1} (a frequency characteristic of α -helical structures), we performed a second derivative analysis¹¹. The second derivative spectra of wild type and 1 month-old Tg animals displayed a peak at around 1640 cm^{-1} , which together with the less intense bands above 1675 cm^{-1} , can be assigned to the intra-molecular β -sheet structures of native proteins^{11,12}. The second derivative spectra of AD transgenic animals are characterised by a peak at \sim 1627 cm^{-1} , due to inter-molecular β -sheet structures. The inter-molecular β -sheet content increased only in AD transgenic and not in wild-type mouse brains at these ages (Fig. 1b-g). This type of intermolecular β -sheet structure has been described as amyloid aggregates in brain tissue samples analysed by μ FTIR^{13,14}.”

Point 3: *The basis for interpretation of the different molecular structural states of A β in the in vivo μ FTIR images is shown in Supplementary Figure 5. The authors should consider a more prominent position for Supplementary Figure 5 (e.g., in the main text).*

Response: We feel that this experiment is preferable as a supplementary experiment mainly because *in vitro*-generated A β 40 oligomers may not have similar structures to oligomers that form *in vivo*, while *in vivo* results are the focus of our study. However, we do also agree that this experiment is important, since it provides an example of β -structured (based on FTIR data), non-fibrillar (based on TEM and SAXS data) oligomers.

Point 4: They should also make it clearer that interpretations are based on FTIR analysis of spectra from the "model compounds" that do not necessarily match structures that would be found *in vivo*.

Response: See also our response to the related Point 3 just above. Further, after re-reading the manuscript with the reviewer's comment in mind, we tried to better acknowledge that the recombinant and synthetic peptides used in our present study as model compounds do not necessarily have to match the *in vivo* structures. We revised our text and added the following sentences (page 5, lines 118-123):

*"These data corroborate previously published experimental data¹⁴ and although the secondary structure of *in vitro*-generated A β oligomers cannot be a true prototype for oligomers that may form *in vivo* due to the complex environment of the brain, these data support the hypothesis that the β -structures detected in brain tissue of Tg19959 mice at 2 months of age may originate from non-fibrillar (non-ThT and non-OC positive) β -sheet A β species formed in brain tissue prior to amyloid plaques."*

Point 5: FTIR measurement of synthetic unstructured A β 40 monomer indicates a peak just above 1640 cm⁻¹. Since A β 40 and A β 42 monomers are generally believed to be unstructured in solution, this is probably a reasonable interpretation. There are some claims in the literature about α -helical content for A β monomers, but alpha-helical structures seem to yield similar peak positions in FTIR. This signature was not observed *in vivo*, suggesting that no unstructured monomers were present.

Response: We agree with this comment and to make the text more clear, we added the following paragraph (page 5, lines 111-123):

*"FTIR measurements of a model compound (synthetic A β 1-40 monomers) showed a peak at 1640 cm⁻¹ (Supplementary Fig. 5a), which corresponds to unstructured A β 40 monomers in the experimental solution (buffered D₂O)¹⁸. This signature was not observed for A β 40 oligomers and fibrils, suggesting that unstructured monomers were not present in the samples. Instead, A β 40 oligomers show a peak at about 1630 cm⁻¹, which indicates the presence of β -sheets¹⁸. However, the electron microscopy, small angle X-ray scattering and ThT spectroscopy data support the non-fibrillar nature of these A β 40 oligomers: lack of thin (12-15 nm), elongated (100-1000 nm) and branched structures¹⁹ (Supplementary Fig. 5 b-d). These data corroborate previously published experimental data¹⁵ and although the secondary structure of *in vitro*-generated A β oligomers cannot be a true prototype for oligomers that may form *in vivo* due to the complex environment of the brain, these data support the hypothesis that the β -structures detected in brain tissue of Tg19959 mice at 2 months of age may originate from non-fibrillar (non-ThT and non-OC positive) β -sheet A β species formed in brain tissue prior to amyloid plaques."*

Point 6: The interpretation of the "oligomer" and "fibril" FTIR signals seems less conclusive. It may be unreasonable to expect perfectly conclusive results from *in vivo* studies like this one, but the paper should at least make it clear what assumptions and conjectures are being made.

Response: We want to point out that our interpretation of "oligomer" and "fibril" is not based on the FTIR data but on the morphology revealed by electron microscopy and small angle scattering. Using FTIR we demonstrate the presence of β -sheet structures in these morphologically non-fibrillar species. To make the text more clear, we have changed the following sentence in the original manuscript:

"However, FTIR also shows the presence of β -sheet structures in non-fibrillar, thioflavin negative synthetic A β oligomers (Supplementary Figure 5 b-d)."

to (page 5, lines 115-118):

"However, the electron microscopy, small angle X-ray scattering and ThT spectroscopy data support the non-fibrillar nature of these A β 40 oligomers: lack of thin (12-15 nm), elongated (100-1000 nm) and branched structures¹⁹ (Supplementary Fig. 5 b-d)."

We are aware that our distinctions are based on a definition of fibrils, which we have now tried to make clearer in the text of our manuscript. The lack of thin (12-15 nm), elongated (100 - 1000 nm) and branched structures on EM is used to indicate that the structures are not fibrils but rather oligomers, that could also be described as globular oligomers. The non-fibrillar nature is further supported by the SAXS data. We are aware that this is our working definition and that it is challenging to fully define structures; we note that reviewer 3 also points out previous evidence for fibrils that are thioflavin and antibody OC negative.

Point 7: *FTIR data on a synthetic Cu-induced A β 40 oligomer exhibits an FTIR peak at 1630 cm⁻¹ (Supplementary Figure 5). It is then assumed that any signal at 1630 cm⁻¹ for the in vivo measurements indicate the presence of an oligomer. FTIR measurement of A β (40) fibrils produced a peak at ~1623 cm⁻¹, leading to the interpretation that signals that this peak position indicate fibrils. However, it is not clear that the position of a single FTIR peak is sufficient to definitively identify oligomer or fibril structure. For example, data in (E. Cerf, R. Sarroukh, et. Al. Biochem. J. 421, 415-423 (2009)), a widely respected paper on FTIR of A β 42 aggregates, indicate that the main FTIR signal from oligomer and fibril β -sheets both show up at 1630 cm⁻¹.*

Response: Indeed, Cerf R et al., 2009, reported that the FTIR signal from oligomer and fibrils may have the same peak position for β -sheets structures. That was the reason why we used electron microscopy and SAXS to study the *in vitro* generated A β oligomers either dried on the surface or in solution, respectively. As in our response to the above Point 6, we note that in the experiments in Supplementary Fig. 5, the interpretations of "oligomer" and "fibril" are based on morphology, which was revealed by electron microscopy and small angle scattering rather than by FTIR. As also noted above, we have revised our text to now make this clearer.

Point 8: *It is known that A β 40 and A β 42 show very different aggregation characteristics (especially regarding oligomers) and it is not clear that the in vitro-generated Cu²⁺ A β 40 oligomers indicate anything about oligomers that may form in vivo.*

Response: We completely agree with this point; it was reported that even the same A β peptide may show structural polymorphism of fibrils (Fändrich M et al., 2008). Since it is not possible to isolate A β of sufficient purity from brain, we turned to synthetic A β 1-40. In our experimental conditions A β 1-40 forms ThT positive amyloid fibrils, and compared to A β 1-42 it does not aggregate as rapidly at concentrations required for biophysical characterization. To further acknowledge that synthetic or recombinant peptides may be different from physiological A β *in vivo*, we added the following text (page 5, lines 119-121):

"...although the secondary structure of in vitro-generated A β oligomers cannot be a true prototype for oligomers that may form in vivo in the complex environment of the brain..."

Point 9: *Furthermore, the BN-PAGE data in Figure 3 indicate that aggregates in the 100-400 kDa range co-exist with tetramers in vivo.*

Response: We agree with the reviewer that in the BN-PAGE in Figure 3 a high molecular weight smear can be seen in the 1-month-old Tg mouse brain. However, we note that we used antibody 82E1 here, which is specific for the N-terminus of A β peptides but also detects APP C-terminal fragments that share the same N-terminus following BACE cleavage. Since the A β 42 C-terminus specific antibody does not see this smear (Fig. 3b), and antibody 369 specific for the C-terminus of APP/CTFs (which does not see A β) does detect the smear, but not the lower molecular weight band, our cumulative data most support that the smear seems not to be A β while the 20 kD band is. However, at and after 2 months of age the smear in Tg mouse brain shifts to being comprised more clearly of A β and less of CTFs (based on APP/CTF bands analysis – Fig. 3c and Supplementary Fig. 6c). It is however also possible that A β is present in this 100-400

kDa range at 1 month of age, which might not be detected by the A β specific antibody due to e.g. antibody sensitivity to the A β conformation (epitope is hidden). We have now re-written text to make this clearer, including adding the following to the revised Figure 3 legend (page 21, lines, 505–513):

“As detected by the human specific 82E1 antibody, human A β (dotted red box) in the Tg19959 mouse brain appears consistent with A β 1–42 tetramers as well as a smear at higher molecular weights. Synthetic human A β 1–42 was used as a size marker and positive control (dotted black box; underexposed). The Western blot shows bands that can be interpreted as 1-, 2-, 3-, 4-mers. (b) BN-PAGE of membrane-associated fractions of mouse brain homogenates at 1 and 2 months of age and subsequent Western blotting with A β 42 specific antibodies 12F4 and MBC42 detect the presence of low molecular weight A β 42 bands (dotted red boxes). The low molecular weight A β 42 band appears specific, since no band is observed in brain tissue homogenate from APP knockout mice.”

Point 10: *What FTIR signatures would one expected from larger non-fibrillar aggregates? In the end, the μ FTIR spectroscopic imaging data seem to indicate that A β structural changes occur in vivo, but it is not obvious how to interpret these changes. Note that the BN-PAGE data support the interpretation that the early aggregates are not fibrils.*

Response: Using FTIR we also examined the high molecular weight fractions of 2-month-old Tg mouse brain, where FTIR spectral analysis revealed the presence of β -sheet structures. However, it was not possible to rule out that this increase might not be secondary to the sample preparation, such as artificial A β aggregation during fractionation of the brain tissue homogenate. There are certainly considerable challenges in defining this β -sheet increase at 2 months in Tg brain by FTIR, which was a reason for why we show the data with synthetic A β despite the issue of these being non-brain preparations. We now modified the following sentence to more clearly point out that the BN-PAGE WBs support the conclusion that the initial A β aggregates are not fibrils (page 6, lines 144–149):

“Overall, these results support the conclusion that under physiological conditions A β in brain initially exists as a low molecular weight protein complex of about 20 kDa. With the age-related increase in A β in AD transgenic mice, this A β complex is then reduced as A β forms higher molecular weight aggregates. Strikingly, this change in the initial state of A β occurs concomitantly with the increase of β -structured content detected by μ FTIR.”

Point 11: *Supplementary Figure 1 is meant to show that the process of freezing and drying does not induce fibril formation (or any other aggregation). FTIR samples show distinct spectral patterns, specifically, the absorbance peak indicating β -sheet structure only occurs in fibrils but not monomers. It is problematic that the statement about the sample preparation is hidden in the middle of the caption, making the figure capture hard to follow [...“frozen and dried (in the same way as the brain tissue slides for μ FTIR imaging)...”].*

Response: We have now presented the sample preparation in Supplementary Figure 1 more clearly and have revised the text of the legend, changed the former sentence and moved the information on the sample preparation from the middle to the beginning of the caption as follows:

“We prepared FTIR samples of A β monomers and fibrils in the same way as the FTIR samples from the brain tissues were prepared. 5 μ L drops of samples corresponding to monomeric and fibrillar fractions were placed on CaF₂ spectrophotometric windows, and then all samples were snap frozen and cryo-dried before measurements were taken.”

Point 12: *Also, it should be emphasized that lack of aggregation in the in vitro preparation (A β 42 fibrils and monomers) does not necessarily indicate that aggregation would not have been affected in the tissue samples. The in vitro and in vivo environments are different and the in vivo environment contains both A β 40 and A β 42 (see the first author's*

own paper, *Biomacromolecules* 2013). While it may be impossible to provide an ideal control experiment for the *in vivo* measurement, this limitation should at least be acknowledged.

Response: We fully agree that ideal standards and controls for the FTIR experiments on brain tissue are a challenge. However, detection of β -sheets in monomeric A β 42 preparations or the degradation of β -sheet A β 42 fibrils in the synthetic samples might have pointed out inappropriate conditions of our sample preparation. We now more clearly acknowledge that the *in vitro* and *in vivo* environments are different in the following revised sentence (pages 3-4, lines 74–78):

“While in vitro and in vivo environments are of course different and ideal controls for the FTIR experiments of brain tissue are challenging, we verified that the sample preparation used for our μ FTIR imaging of cryo-dried brain tissue did not introduce artificial β -sheet formation or that the loss of β -sheet did not occur in synthetic A β 42 preparations (Supplementary Fig. 1).”

Point 13: *In Kelementeva's (the first author) Biomacromolecules paper, she used the same AD Tg mice and study the A β aggregation, it seems the A β 42/A β 40 ratio was increased in the 5-month-old Tg mice and there is still a strong 20kDa band shown in the non-SDS PAGE, this is not consistent with the current paper, which claims the 20kDa A β 42 protein complex will be eliminated as the Tg mice aging. Could this apparent contradiction be explained?*

Response: We note that the experimental animals tested in this 2013 study were APP/PS1 mutant transgenic mice, while in the present study we used Tg19959 mice, which have two APP but no PS1 mutations and develop amyloid plaques and cognitive decline at an earlier age. In addition, the NB PAGE shown in Fig. 6 (I) in the study from 2013 used the 100 000g soluble fraction of brain homogenates (TBS-soluble fraction) blotted with antibody 6E10, which is specific to full-length APPs, APP CTFs and A β peptides, while in the present study we used 14 000g membrane soluble fractions blotted with C-terminus specific A β 42 antibodies. Taken together we cannot directly compare the results from the 2013 and 2016 studies. However, in retrospect, the 2013 results could support our conclusion that in early stages A β exists as a 20 kDa complex, although we did not comment on this band in the 2013 study.

Point 14: *The authors conducted experiments to show co-localization of A β 42 and APP. This would appear to be a notable result. However, little is offered in the paper regarding interpretation of this finding. Why is it significant that A β interacts with APP?*

Response: We agree that this point was not sufficiently discussed. Since the increase of A β is concomitant with a change in APP structure, and since *in vitro* experiments suggest that A β may bind APP, we now mention (page 8, lines 210-213) the possibility that A β binding to APP may cause APP structural changes that might interfere with the physiological processing of APP:

“Further, we present evidence consistent with aggregation of APP concomitant with A β pathogenesis. Since the increase of A β is concomitant with an apparent change in APP structure, and since in vitro experiments support that A β may bind APP, it is possible that binding with A β may cause changes in APP structure that interfere with physiological processing of APP.”

Point 15: *The sentence “Thus, the current study supports the novel concept that also in AD destabilization of native conformations of proteins linked pathologically and genetically to the disease might be critical for this most common neurodegenerative disease, and that AD may, therefore, be similarly amenable to treatment by stabilization of normal protein complexes.” on page 8 is too long and not clear.*

Response: Based on this and another reviewer's (reviewer 1, point 7) comments, we have significantly changed this former sentence (and others in the Discussion) in the

original manuscript to reduce our former emphasis on this more speculative part, including on page 9, lines 220-224:

"Thus, the current study supports the novel concept that also in AD the native and physiological conformations of proteins linked pathologically and genetically to the disease might be altered in the disease. Stabilisation of the physiological structure might therefore be considered among novel therapeutic approaches also for this most common neurodegenerative disease."

Reviewer 3

We also thank reviewer 3 for the valuable input and interest in our study expressed as e.g.: *"The results are generally original and interesting"*.

Reviewer 3 made specific comments focused on ThT and antibody OC specific fibril detection, and on FTIR spectral analysis regarding beta-parallel/anti-parallel content. In response to this reviewer's comments, we modified the manuscript by adding more of an explanation on the FTIR data and revised our Discussion.

Point 1: *P4, paragraph1: "prior to amyloid plaque formation as determined by thioflavin S (ThS) staining". Thioflavin S staining is not a reliable definition of a plaque even though it has been used as an operational definition for a number of years. There is increasing evidence that it is just another probe of amyloid conformational differences. Some fibrils bind it; other fibrils don't. Some oligomers bind it; others don't. There are many published examples of this, but the most recent one that sticks in my mind was published this past week on the Osaka mutation A β fibrils by Matthew Elkins and Mei Hong from MIT in JACS. Similarly, it is known that OC reacts with beta sheet aggregates that are oligomeric and not morphologically fibrillar. This does not challenge the author's observation of conformational differences, but only the claim that deposits that are not ThS positive are not fibrils or plaques or that OC positive deposits are necessarily fibrillar.*

Response: We appreciate these insightful comments by the reviewer and based on these, added the JACS reference on page 5, line 110, and also changed the following sentence (page 5, lines 121-123):

"...these data support the hypothesis that the β -structures detected in brain tissue of Tg19959 mice at 2 months of age may originate from non-fibrillar (non-ThT and non-OC positive) β -sheet A β species formed in brain tissue prior to amyloid plaques."

As discussed also in our responses to Reviewer 2 we now provide a clearer definition of how we define fibrils, which is based on EM morphology, SAXS and dye/antibody labelling, page 5, lines 117-118: *"lack of thin (12-15 nm), elongated (100-1000 nm) and branched structures..."*

Point 2: *Vincent Raussens has shown that parallel and antiparallel beta sheet amyloids can be distinguished by their 1692.1639 ratio of their FTIR spectra and he has speculated that this may form the difference between antiparallel, oligomeric forms that react with conformation dependent antisera, A11 and parallel, fibrillar forms that react with OC sera (Biochem J. 2009 Jul 15;421(3):415-23. doi: 10.1042/BJ20090379.). Can you interpret your FTIR spectra in terms of parallel and antiparallel beta sheet content? Do the 2 month deposits that do not stain with OC, stain with A11?*

Response: We now analysed the 1695 cm^{-1} band as described in Cerf et al., 2009.. We did not find any significant changes in the intensity of 1695 cm^{-1} bands between 1 and 2 month-old animals. However, it is possible that changes in orientation of β -strands may be more difficult to appreciate in tissue compared to the purer *in vitro* preparations studied by Cerf et al., since such features could be hidden by, among other reasons, the heterogeneity of brain tissue and the presence of numerous other proteins. We also agree that an antibody against more soluble oligomers, such as A11, would be highly interesting, but as noted also in our response to reviewer 2, we have had a difficult time with specific labelling using antibody A11 in our mouse brain sections. Actually, over the

past decade several different investigators in our group have not been able to obtain clear labelling in mouse brain sections with this antibody. As the reviewer will know, so many factors can be responsible for this, and we did also try again following the current reviews, but again could not see clear labelling. Please also see our response to reviewer 1, point 2.

Minor point 1: *Introduction: It isn't necessary to demean other approaches to validate the approaches used here and this verbiage may not endear the authors to practitioners of alternative approaches who may question the "non-destructive" nature of synchrotron IR radiation.*

Response: We also appreciate this comment and did not want to demean other approaches, and have now edited our manuscript accordingly. We can mention that the synchrotron IR radiation used in our experiments is not considered to be destructive, since mid-IR photons are too low in energy (0.05–0.5eV) to either break chemical bonds or to cause ionisation. This lack of damage is also described by among others: Diem M., et al. A decade of vibrational micro-spectroscopy of human cells and tissue (1994-2004). *Analyst*. 129(10):880-5 (2004), Holman Y.N. et al., Synchrotron-Based FTIR Spectromicroscopy: Cytotoxicity and Heating Considerations *H.-J Biol Phys.* 29(2-3): 275–286 (2003), and Chan K. L. and Kazarian S.G. Attenuated total reflection Fourier-transform infrared (ATR-FTIR) imaging of tissues and live cells. *Chem. Soc. Rev.*, 45, 1850-1864 (2016). We now comment on this and added these citations in the text (page 3, lines 69–70):

" μ FTIR is a non-destructive technique since, mid-IR photons are too low in energy (0.05–0.5eV) to either break chemical bonds or to cause ionization^{6,7}."

In addition, we now note the sensitivity of the method (page 3, lines 68– 69):

" μ FTIR permits the acquisition of spectra from samples as low as 100 pg⁵;"

We also note that the sample preparation for FTIR does not include the following common steps such as separation, purification and concentration, procedures that may result in the loss of material or undesired change in structure of A β . However, we also edited the former sentence that seemed most negative about other methods to the following (page 3, lines 49–52):

"Studies on the aggregation state of A β in brain typically have relied on brain tissue being processed by among others, homogenization, high-speed centrifugation and/or enrichment, all of which may trigger alterations in protein structure and state of assembly of aggregation prone proteins."

Minor point 2. *Similar high molecular weight A β was also observed on BN-PAGE in aged APP23 AD transgenic.*

Response: As we reference, Upadhaya et al., 2012, did also note that with age A β forms higher molecular weight aggregates although in other AD transgenic mouse models (APP23 and APP51/16 tg mice). However, the high molecular smear prior to plaques was only very faintly seen in one of their mouse models (APP23). Because this smear was so weak the authors had doubts about this labelling that they expressed as follows: *"...under native conditions by BN-PAGE revealed no soluble and dispersible A β 42 and A β 40 in 5-month-old animals except for a 240–480 kDa, very weakly A β 42-positive smear in APP23 mice (Fig. 10A). We can neither exclude that this smear may represent an unspecific result nor that it indicates the first sign of soluble A β oligomers in these animals."*

Together with our data, these results may be interpreted as the first sign of soluble A β oligomers. In addition, we note that our methods also differed: we did not use sucrose in our homogenization buffer (since sucrose environments can affect A β aggregation - unpublished data) and we did not sonicate brain homogenates. In addition, our blots were developed with enhanced chemiluminescence (SuperSignalR West Femto) detection system (Thermo Fisher Scientific) with up to 1000 sec exposure time.

REVIEWERS' COMMENTS:

Reviewer #1 (Remarks to the Author):

I have carefully reviewed the changes made to the manuscript in response to my reviewer comments and I am satisfied that all the comments have been fully addressed. It was very helpful that the authors provided a response to the comment regarding the possible use of A11. I believe that the paper is now improved and I recommend publication in the current form.

Reviewer #2 (Remarks to the Author):

I am satisfied that the authors have carefully considered and addressed the comments made by myself and the other reviewers. With in vivo studies like this one, it is nearly impossible to draw direct conclusions about molecular structure, as spectral signatures and antibody binding (we have also experienced unpredictable behavior of A11) cannot be directly compared between in vitro and in vivo experiments. The authors have done a good job of explaining the assumptions behind their interpretations, and connecting these assumptions to previous research.

There does seem to be a minor misunderstanding between me and the authors, regarding the interpretation of FTIR peak positions. Nevertheless, the authors have (perhaps inadvertently) addressed my concern in their new revisions to the text. I never meant to question the interpretation of monomeric vs. oligomeric vs. fibrillar structure for the model compounds. I realize that electron microscopy and X-ray scattering measurements were performed on the in vitro generated samples and provide a clear basis for structural interpretation. I was questioning whether one could interpret similarity of FTIR peak positions from in vivo samples to indicate structural similarity. As I stated, the manuscript text in its present form appropriately emphasizes this ambiguity.

Reviewer #3 (Remarks to the Author):

All of my comments have been adequately addressed in the revised manuscript.
Charles Glabe

Reviewer 1 wrote *“I have carefully reviewed the changes made to the manuscript in response to my reviewer comments and I am satisfied that all the comments have been fully addressed. It was very helpful that the authors provided a response to the comment regarding the possible use of A11. I believe that the paper is now improved and I recommend publication in the current form.”*

Reviewer 2 wrote: *“I am satisfied that the authors have carefully considered and addressed the comments made by myself and the other reviewers.”* and then added: *“There does seem to be a minor misunderstanding between me and the authors, regarding the interpretation of FTIR peak positions. Nevertheless, the authors have (perhaps inadvertently) addressed my concern in their new revisions to the text. I never meant to question the interpretation of monomeric vs. oligomeric vs. fibrillar structure for the model compounds...”* We believe that we understood the main point the reviewer had made about the challenge of applying *in vitro* results to the brain tissue situation, which we highlighted in our revision, but can also understand why the reviewer felt that we might have misunderstood, since we see that we extended our responses to the reviewer (specifically our response to point 7 of Reviewer 2) to a topic (FTIR differences between synthetic oligomers and fibrils) that the reviewer was not emphasizing in this point. Reviewer 3 noted: *“All of my comments have been adequately addressed in the revised manuscript.”*

We also note here that we expanded our background and discussion as we were asked to, moved former supplementary 5 to the main text as was also suggested by reviewer 2 in the initial review, and added reference to Cerf et al. 2009 that both reviewers 2 and 3 had mentioned in their initial reviews.